# Massive peatland carbon banks vulnerable to rising temperatures

A. M. Hopple [1,2,3,7 ✉], R. M. Wilson[4,7], M. Kolton [5], C. A. Zalman[2], J. P. Chanton[4], J. Kostka [5], P. J. Hanson [6], J. K. Keller[2] & S. D. Bridgham[1]

Peatlands contain one-third of the world's soil carbon (C). If destabilized, decomposition of this vast C bank could accelerate climate warming; however, the likelihood of this outcome remains unknown. Here, we examine peatland C stability through five years of whole-ecosystem warming and two years of elevated atmospheric carbon dioxide concentrations (eCO$_2$). Warming exponentially increased methane (CH$_4$) emissions and enhanced CH$_4$ production rates throughout the entire soil profile; although surface CH$_4$ production rates remain much greater than those at depth. Additionally, older deeper C sources played a larger role in decomposition following prolonged warming. Most troubling, decreases in CO$_2$:CH$_4$ ratios in gas production, porewater concentrations, and emissions, indicate that the peatland is becoming more methanogenic with warming. We observed limited evidence of eCO$_2$ effects. Our results suggest that ecosystem responses are largely driven by surface peat, but that the vast C bank at depth in peatlands is responsive to prolonged warming.

[1] Institute of Ecology and Evolution, University of Oregon, Eugene, OR 97403, USA. [2] Schmid College of Science and Technology, Chapman University, Orange, CA 92866, USA. [3] Pacific Northwest National Laboratory, Richland, WA 99354, USA. [4] Earth, Ocean, and Atmospheric Sciences, Florida State University, Tallahassee, FL 32306, USA. [5] School of Biological Sciences and School of Earth and Atmospheric Sciences, Georgia Institute of Technology, Atlanta, GA 30332, USA. [6] Environmental Sciences Division, Oak Ridge National Laboratory, Oak Ridge, TN 37831, USA. [7] These authors contributed equally: A. M. Hopple, R. M. Wilson ✉email: anyahopple@gmail.com

Peatland soils represent a major global carbon (C) stock that is sensitive to climate change[1–3]. Increases in temperature and atmospheric carbon dioxide ($CO_2$) concentrations, along with corresponding changes in hydrology, have the potential to stimulate the return of stored soil C to the atmosphere as $CO_2$ and/or methane ($CH_4$)[4–8], amplifying the drivers of climate change. However, prior soil-warming experiments have primarily focused on surface responses to warming and long-term data from manipulative field studies remain scarce[8–10]. Therefore, the fate of peatland catotelm C under future climate conditions remains largely unknown.

The SPRUCE project alleviates this knowledge gap with an in-situ, manipulative experiment that addresses climate-driven questions on an ecosystem scale over the span of a decade[11]. Located in an ombrotrophic peatland in northern Minnesota, USA (S1 Bog)[12], this study uses a regression-based experimental design that warms the vegetation and peatland soil profile to a depth of 3 m within ten, 12-m-diameter enclosures to five target temperature differentials (+0 to +9 °C above ambient), with duplicate enclosures subjected to ambient and ~+500 p.p.m.v. above ambient atmospheric $CO_2$ concentrations (e$CO_2$; Supplementary Fig. 1a). Capturing comprehensive peatland responses to climate change requires the use of active surface and deep-soil warming because deep-soil temperatures will naturally increase in concert with rising annual air temperatures[13–15]. Based on current greenhouse gas emission rates[16], forcing estimates in line or greater than the RCP 8.5 scenario are likely to occur. This will result in 2.6–4.8 °C increases in global mean surface temperatures by 2100[17], with Minnesota experiencing 4.07–4.14 °C increases in annual air temperature[18]. However, both models and observations indicate that northern boreal forests and tundra will continue to be exposed to greater warming than most other terrestrial biomes. This is largely due to positive feedback effects related to decreases in surface albedo resulting from decreasing sea ice and shorter periods of winter snow cover[17]. The effects of temperature increases are further compounded by the heightened frequency of extreme heat events which are expected to expose peatlands to acute heat stress, exceeding the conditions for which vegetation is currently adapted. Thus, under a globally averaged projection of +4 °C warming by the end of this century, boreal regions may experience temperature increases up to 8.3 ± 1.9 °C[19,20]; thus, the +9 °C treatment implemented in this study represents an upper limit on what can be expected under the most extreme scenarios. In addition, implementing warming treatments at a range of temperatures allows for the exploration of ecosystem-wide response surface thresholds and nonlinear curve response fitting[21].

Whole-ecosystem warming (WEW) was initiated in August 2015, following 14 months of deep-peat heating (DPH). During the DPH phase of this experiment, deep-soil temperature targets were successfully maintained throughout the year; however, the lack of air warming resulted in muted surface warming[11,22]. After the introduction of air warming, target temperature differentials were attained from the atmosphere to peat depths of at least 2 m[11]. Finally, e$CO_2$ was introduced in a subset of the enclosures in June 2016 (Supplementary Fig. 1b).

Here, we present four years of data (2015–2018) that expand upon early results from the SPRUCE experiment[22] by exploring the response of peatland $CH_4$ cycling through 5 years of warming and 2 years of e$CO_2$. We continue to observe exponential increases in surface $CH_4$ emissions and although surface $CH_4$ production rates remain much greater than those at depth, rates of $CH_4$ production are now positively responding to increasing temperatures throughout the entire soil profile. In addition, while $CH_4$ was produced primarily from decomposition of modern surface photosynthate prior to warming, radiocarbon analyses indicate that older deeper C sources are playing a larger role following prolonged warming. Most troubling, we observe decreases in the ratios of $CO_2:CH_4$ in gas production, porewater concentrations, and emissions, providing three lines of independent evidence that the peatland is becoming more methanogenic with warming. Finally, while we observed limited evidence of any e$CO_2$ effect to date, it remains to be seen whether this treatment will eventually modify the observed temperature effects via future ecological cascades. Our current results suggest that ecosystem responses remain largely driven by surface peat, but that following a relatively short lag, the vast C bank at depth in peatlands is responsive to warming.

## Results and discussion
**Ecosystem responses to prolonged WEW and e$CO_2$.** During WEW, increasing temperatures stimulated $CH_4$ production rates

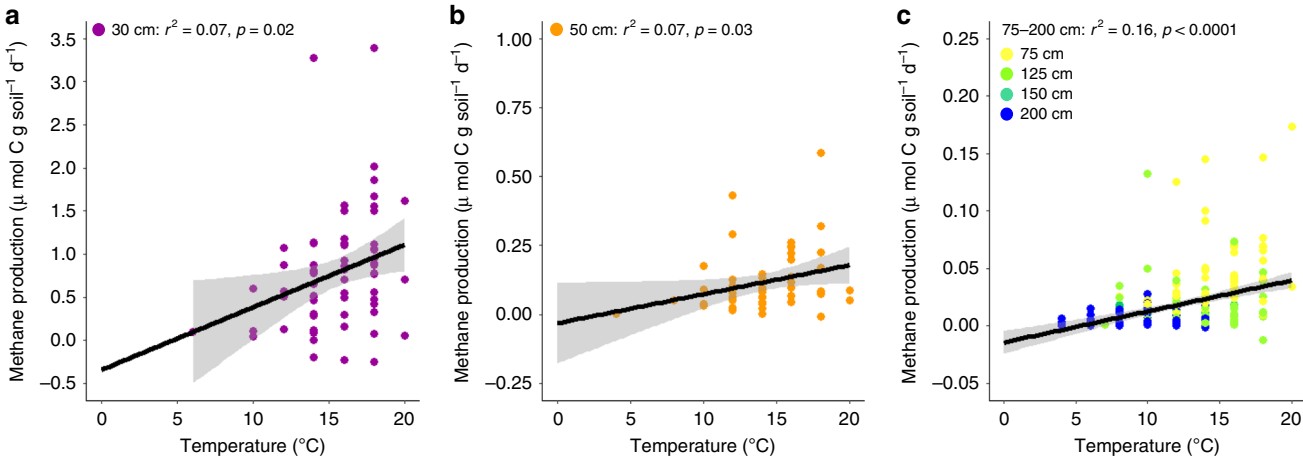

**Fig. 1 Depth-specific $CH_4$ production from anaerobic incubations.** Positive $CH_4$ production temperature responses from peat samples collected (**a**) 30 cm, (**b**) 50 cm, and (**c**) 75–200 cm bellow the hollow surface at S1 Bog and anaerobically incubated within 1 °C of in-situ temperatures. Depth increments are separated into statistically different groups due to a significant interaction between temperature and depth ($p < 0.0001$). Peat samples were collected 1–4 times per year during the growing season, and over the course of 4 years (2015–2018) throughout whole-ecosystem warming. Linear regressions with 95% confidence intervals are shown in black and gray, respectively. Colors represent different depth increments. A 50-cm outlier was excluded from (**b**); however, when this point was included the linear regression remained significant ($r^2 = 0.10$, $p < 0.0001$). Note differences in the $y$-scales among the panels.

throughout the peat profile (30–200 cm below the hollow surface; $p \leq 0.05$; Fig. 1), as well as $CO_2$ production rates of deep peat ($\geq 75$ cm; $p < 0.01$; Supplementary Fig. 2) in anaerobic laboratory incubations at in-situ temperatures. This contrasts with earlier results which showed no temperature effect on $CH_4$ and $CO_2$ production in catotelm peat ($\geq 50$ cm) following 14 months of DPH[22]. It is possible that this lag period may be longer under natural conditions where the soil profiles warms from the top down.

A positive temperature response at depth does not in and of itself indicate enhanced decomposition of ~8000–10,000-year-old soil C. However, comparisons of the radiocarbon ([14]C) content of dissolved inorganic carbon (DIC = $CO_2$) and $CH_4$ (decomposition products) with those of DOC and solid peat (decomposition sources) allowed us to discern whether the organic matter fueling heterotrophic decomposition was recent photosynthate (DOC) or ancient peat C[23–26]. While $CH_4$ [14]C analyses were not conducted at every depth and timepoint, multiple studies have shown that the [14]C contents of $CO_2$ and $CH_4$ are tightly coupled[23] (Supplementary Fig. 3). Previous research has shown that almost all heterotrophic decomposition in deep peat (up to 2 m) at S1 Bog was driven by relatively young, surface-derived DOC prior to WEW[22,26]. However, during WEW, porewater DIC, and presumably $CH_4$, appear older ([14]C depleted) in the warmest enclosures when compared to the decomposition products of ambient temperature plots (Fig. 2). While the [14]C signatures in the warmest treatment plots remain much younger than that of solid peat[22], this does suggest that older peat C has begun to play a larger role in decomposition at depth following warming; however, the decomposition of older C at depth took years to be observed.

We initially hypothesized that this delayed response may be due to the slow growth of methanogens given the low thermodynamic yield of methanogenesis under in-situ conditions[27,28]. Yet, the abundance of methanogens in peat (using quantitative PCR of the *mcrA* gene) did neither change with temperature ($p = 0.59$; Supplementary Fig. 4) nor with increased time exposed to warming ($p = 0.58$) during WEW. We hypothesize alternative microbial community attributes, likely physiological activity and/or composition, must be underlying observed increases in deep $CH_4$ production. For example, recent experiments suggest that the temperature optimum for S1 Bog methanogens is around 30 °C[29]. Thus, the warmer treatments likely stimulated methanogen activity as soil temperatures were closer to their thermal optimum. Furthermore, laboratory-based incubation experiments using S1 Bog peat linked increases in $CH_4$ production to rising temperatures, as well as shifting methanogen community composition[29].

**Lagging ecological cascades.** While warmer temperatures enhanced $CH_4$ production at depth in S1 Bog following a lag period, surface (to 50 cm depth) $CH_4$ production rates remained much greater than those at depth ($p < 0.0001$), accounting for 75% of total $CH_4$ production (Supplementary Table 1). However, temperature explained much less of the variation in surface $CH_4$ production during WEW ($r^2 = 0.07$, Fig. 1) relative to DPH ($r^2 = 0.71$)[22]. Climate-induced perturbations to the ecosystem, such as changes in water table, increased belowground exudation of labile plant compounds, or changes in plant and/or microbial community composition, likely had cascading ecological effects on peatland $CH_4$ production (Updegraff et al.[8], Turetsky et al.[9]), muting the direct effects of temperature on this process in surface soils. For example, water-table drawdowns observed under WEW[11] (Supplementary Fig. 5) likely oxidized terminal electron acceptors and intermittently decreased the soil anaerobic zone, suppressing rates of surface methanogenesis and potentially stimulating $CH_4$ oxidation. Conversely, 2 years of exposure to $eCO_2$ stimulated rates of $CH_4$ production in surface soils (Supplementary Fig. 6), possibly through increased availability of the methanogenic substrates, acetate and $H_2$, due to enhanced plant-root exudation[6,7]. We observed decreases in these substrates throughout the entire soil profile during WEW (Supplementary Figs. 7 and 8), indicating heightened microbial substrate utilization with warming. In addition, this negative temperature response became stronger for surface acetate concentrations following the introduction of $eCO_2$ ($p < 0.05$; Supplementary Fig. 9). While temperature explains a small, but statistically significant

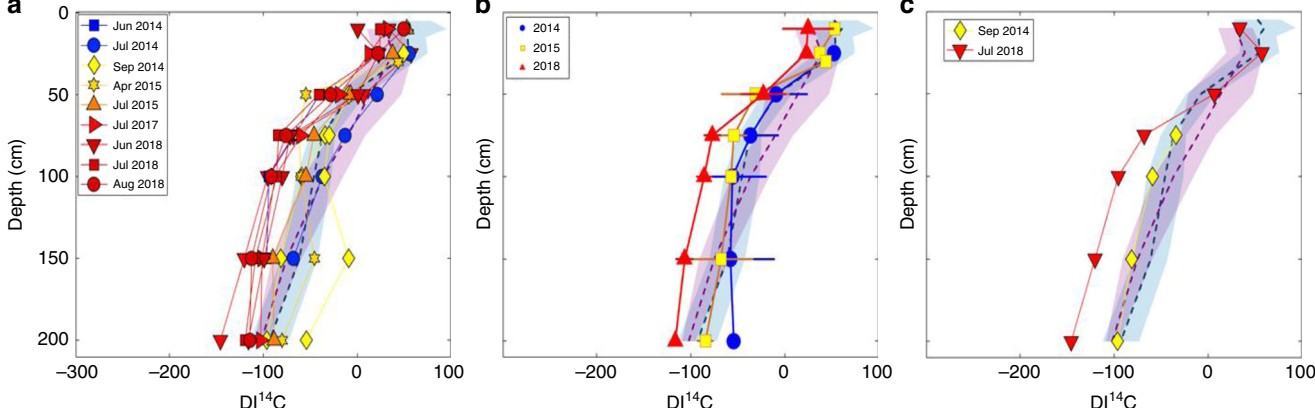

**Fig. 2 Radiocarbon measurements of decomposition products.** Depleted DI[14]C signatures (in ‰) from (**a**) three of the warmest enclosures (+6.75 °C with ambient $CO_2$, + 9 °C with ambient $CO_2$, and +9 °C with elevated $CO_2$) following 5 years of warming. Due to a possible effect of elevated $CO_2$ on DI[14]C signatures (see "Methods"), we also provide [14]C depth profiles for the (**b**) +9 °C and (**c**) +6.75 enclosures without elevated atmospheric $CO_2$ concentrations during 5 years of warming. Symbols and colors denote individual samples collected within each experimental enclosure from 2014 to 2018. The 2014 data points were during deep-peat heating, prior to the initiation of whole-ecosystem warming. Shaded areas and dotted lines indicate the DI[14]C LOESS locally weighted polynomial regression smooth curves, 95% confidence intervals from one ambient temperature enclosure (pink shading), and one reference plot with no infrastructure (blue shading). Control samples were collected over the same time period. In (**b**), symbols and whiskers indicate averages ± 1 standard deviation of each year for samples collected from the +9 °C with ambient $CO_2$ enclosure. Three samples from each depth were collected in 2018 and plotted, but at some depths the standard deviation was much smaller than the symbols at this resolution. As a reference, the solid peat matrix has a [14]C signature that ranges from −400 to −600‰ at depths below 100 cm[6].

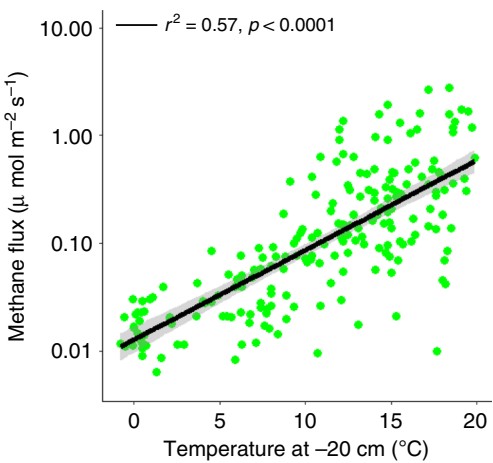

**Fig. 3 Methane flux from S1 Bog.** Exponential temperature response of surface $CH_4$ emissions following five years of warming. Measurements were made 6–8 times per year during 2015–2018. Linear regressions with 95% confidence intervals are shown in black and gray, respectively. Note the log-scale on the y-axis.

proportion of the variance in methanogenic substrate concentrations (acetate: $r^2 = 0.02$–$0.17$, $p < 0.0001$ to $0.08$; $H_2$: $r^2 = 0.02$, $p < 0.01$), our ability to connect rapidly utilized, low-molecular-weight microbial substrates with process rates, and, ultimately, ecosystem function remains a powerful approach for understanding ecological cascades. Together, these results suggest that warming and $eCO_2$ drove increases in surface $CH_4$ production through kinetic- and substrate-based effects that enhanced rates of methanogen activity, overwhelming any suppression effects associated with water-table fluctuations.

**Surface gas emissions.** Increases in $CH_4$ and $CO_2$ production rates throughout the soil profile resulted in exponential and linear increases in peatland $CH_4$ ($p < 0.0001$; Fig. 3) and dark $CO_2$ ($p < 0.0001$; Supplementary Fig. 10) emissions, respectively. Exponential increases in $CH_4$ emissions were also observed during DPH[22] and have persisted over the past 4 years, despite general decreases in water-table position with warming (Supplementary Fig. 5). AIC-based model selection found temperature, water-table position, and their interaction to be the best predictors of $CH_4$ flux, explaining 45% of its variation. The persistent exponential increase in $CH_4$ emissions with warming, despite water-table drawdown, is consequential to positive climate feedbacks.

**$CO_2$:$CH_4$ ratios—the flow of C through anaerobic ecosystems.** Finally, we found decreases in the $CO_2$:$CH_4$ ratios of production, porewater concentrations, and emissions following 5 years of warming as a result of increased $CH_4$ production (Fig. 4). The $CO_2$:$CH_4$ ratio is an ecologically meaningful parameter because in anaerobic systems, organic C is converted to either $CO_2$ or $CH_4$ as the final product, depending on the pathway of decomposition; thus, $CO_2$:$CH_4$ ratios indicate whether the flow of C through an ecosystem has been altered as climate changes. During WEW, $CO_2$:$CH_4$ production ratios decreased exponentially with warmer temperatures across all depth increments ($p < 0.0001$; Fig. 4a). Similarly, porewater $CO_2$:$CH_4$ ratios linearly decreased with warming ($p < 0.0001$; Fig. 4b), although the severity of this effect decreased with depth ($p < 0.001$). There was higher variability in the shallowest (10 cm) depth as evidence by the larger spread of 95% confidence intervals which likely resulted from fluctuations in the water-table depth in the near surface. In addition, while emission $CO_2$:$CH_4$ ratios were stable during the first 2 years of

WEW (2015 and 2016), this ratio began to decrease in 2017 ($p < 0.01$), with an even steeper decrease observed in 2018 ($p < 0.001$; Fig. 4c). While porewater concentration and emission measurements encompass both autotrophic and heterotrophic respiration, we found that the declining $CO_2$:$CH_4$ ratios in both the porewater and the emissions were caused by increasingly methanogenic heterotrophic soil respiration (rather than decreasing $CO_2$ rates). Together, these three lines of independent evidence suggest that the ecosystem as a whole is becoming more methanogenic. This is troubling because $CH_4$ is a potent greenhouse gas with 45 times the sustained-flux global warming potential of $CO_2$ over a 100-year timeframe[30]. Thus, even if warming stimulates plant biomass production and enhances soil C sequestration, these effects are unlikely to completely offset the climate forcing due to increased $CH_4$ emissions.

It should be noted that we report the warming response of one bog and $CH_4$ emissions and anaerobic C cycling have been shown to vary substantially among southern boreal habitats[31,32]. Thus, the results described here do not necessarily reflect the expected or observed responses from other peatland habitats. Nonetheless, our conclusions have far-reaching implications for predicting ecosystem-atmosphere feedbacks that exist in systems experiencing climate change, highlighting the need for similar manipulative studies implemented across a diverse array of ecosystem types and biomes. Finally, as the SPRUCE experiment continues into the next decade, it remains to be seen whether the observed temperature effects will persist, diminish through acclimation of the ecosystem[33,34], or be further modified by the impacts of changes in water-table position and $eCO_2$ on plant-community productivity and composition. However, our results indicate that the vast stores of C at depth in peatlands are vulnerable to rising temperatures, but that ecosystem responses remain largely driven by surface peat and, together, these responses have resulted in a more methanogenic peatland.

## Methods

**Site description.** The Spruce and Peatland Responses Under Changing Environments (SPRUCE) experimental site (http://mnspruce.ornl.gov/), S1 Bog, is an 8.1 ha peatland in north-central Minnesota, USA within the US Forest Service Marcell Experimental Forest (N 47°30.476′; W 93°27.162′). Since the 1960s, extensive scientific investigations have been done at this site and include in-depth descriptions of its physicochemical and biotic characteristics[14,22–26,29,35]. This precipitation-fed, ombrotrophic bog has a perched water table with an average pH of 4.1 at the surface which increases with depth to roughly 5.1 at 2 m depth. The overstory vegetation is primarily dominated by *Picea mariana* (black spruce) and secondarily by *Larix laricina* (larch), while the understory is composed of low ericaceous shrubs, such as *Rhododendron groenlandicum* (Labrador tea) and *Chamaedaphne calyculata* (leatherleaf), and herbaceous perennials, such as *Maianthemum trifolium* (three-leaved Solomon's seal) and *Eriophorum vaginatum* (cotton grass). The bog surface is characterized by hummock and hollow microtopography, with a typical relief of 10 to 30 cm between the tops of the hummocks and the hollows. *Sphagnum magellanicum* generally colonizes the hummocks, while *S. angustifolium* and *S. fallax* cover the hollows. The belowground peat profile and geochemistry are described in ref. [26].

**Whole-ecosystem warming and elevated atmospheric [$CO_2$].** The SPRUCE project uses a regression-based experimental design that warms the vegetation and peatland soil profile to 3 m depth within ten, 12-m-diameter enclosures to five target temperature differentials (+0, +2.25, +4.5, +6.75, and +9 °C), with duplicate enclosures subjected to ambient and ~ +500 p.p.m.v. atmospheric $CO_2$ concentrations ($eCO_2$; Supplementary Fig. 1a). Whole-ecosystem warming (WEW) is achieved within open-topped enclosures (7 m tall by 12.8 m in diameter) by combining air and belowground warming. Air is warmed with propane heaters, whereas belowground warming is attained using low-wattage, 3-m deep, below-ground concentric rings of heaters[11]. The open-top enclosure design allows for surface air warming and enhancement of atmospheric $CO_2$, while subsurface corrals hydrologically isolate each experimental enclosure and allow for changes in water-table level associated with climate manipulation to occur.

Whole-ecosystem warming was initiated 12 August 2015, following 14 months of deep-peat heating (DPH). During the DPH phase of this experiment, deep-soil temperature targets were successfully maintained throughout the year following a gradual treatment equilibration period (~3 months); however, the lack of air

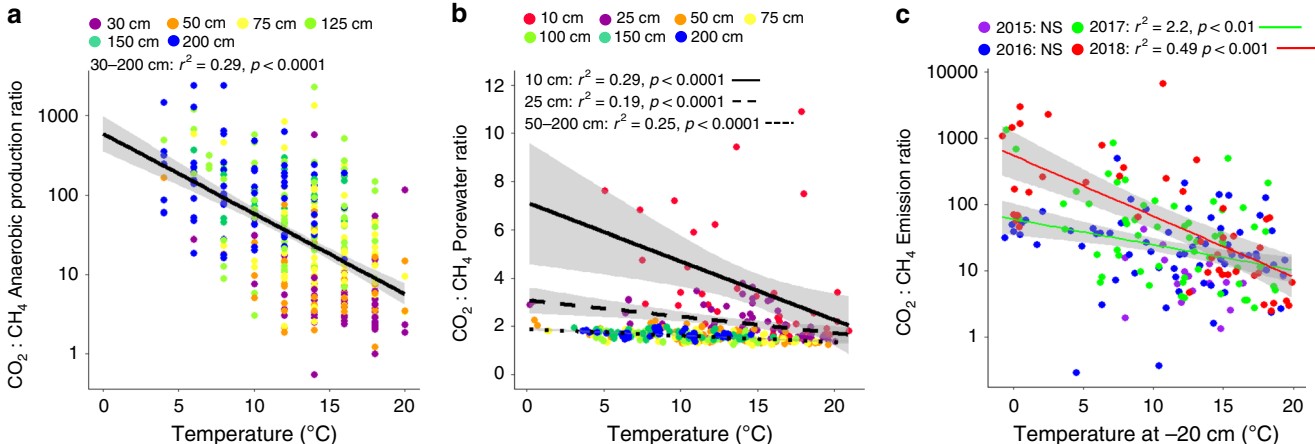

**Fig. 4 $CO_2$:$CH_4$ production, porewater concentration, and emission ratios.** Decreasing $CO_2$:$CH_4$ ratio temperature responses from peatland **a** depth-specific $CO_2$ and $CH_4$ production rates from anaerobic incubations, **b** depth-specific in-situ $CO_2$ and $CH_4$ porewater, and **c** surface $CO_2$ and $CH_4$ emissions following 5 years of warming. Emission measurements were collected 6–8 times per year during 2015–2018. Production measurements were made using peat samples collected from six depths during the same time period and anaerobically incubated within 1 °C of in-situ temperatures. Porewater samples were collected from piezometer wells 3–4 times per year in 2016 and 2017. In (**a**) and (**b**), colors represent different sampling depths and in (**c**), colors represent different sampling years. Linear regressions with 95% confidence intervals are shown in black and gray, respectively. NS not significant.

warming resulted in reduced temperature separation among treatments at the surface[11]. After the introduction of air warming (which signaled the start of WEW), we attained 9 °C temperature separation and differentials across treatment enclosures from the tops of the trees to peat depths of at least 2 m. Temperature differentials have largely been maintained thought the WEW period, with some variation observed in surficial peat zones due to rain and snow events[11]. Finally, $eCO_2$ was introduced in a subset of the enclosures on 15 June 2016, completing the full set of experimental climatic manipulations planned by the SPRUCE project (Supplementary Fig. 1b). In these enclosures, ambient atmospheric $CO_2$ concentrations were on average elevated by +500 p.p.m.v. using pure $CO_2$ from a fossil source (i.e., $^{14}C$-free $CO_2$). The mixture of local ambient air (+0 to 9‰) with pure $CO_2$ yielded $eCO_2$ chambers having typical values of −520 to 540‰ on the $\Delta^{14}C$ scale. Due to the compounding effect of the treatment, we include all 10 enclosures in most analyses and explicitly explore $eCO_2$ in some cases.

**$CH_4$ and $CO_2$ production.** Following the same protocol that was used throughout the DPH experimental phase[22], intact soil cores were collected from multiple depths within each experimental enclosure to discern how rates of $CH_4$ and $CO_2$ production, as well as $CO_2$:$CH_4$ ratios, varied with climate treatment and depth. Sampling events were conducted 1–4 times per year during the growing season, and over the course of 4 years (2015–2018) throughout WEW. In 2015, soil cores were collected from 20 to 30, 50 to 75, 100 to 125, 125 to 150, and 175 to 200 cm depth increments (depth increments are denoted with the lower end of their ranges in figures). We used the same sampling protocol from 2016 to 2018, but collected soil cores at 40–50 cm instead of 125–150 cm to better capture variation in surficial peat horizons. All depths were measured relative to the surface of the hollows. To prevent compression of surface peat samples, a serrated knife was used to collect a 10 cm diameter core from the hollow surface to ~20 cm within the peat profile. A 5-cm diameter Russian corer was subsequently used to extract the remaining samples up to 2 m deep. Soil cores were immediately flushed with nitrogen ($N_2$) in the field to minimize exposure to aerobic conditions. In addition, porewater samples were anaerobically collected from within each enclosure using 1.25 cm diameter PVC piezometers installed at corresponding depth increments (25, 50, 75, 100, 150, and 200 cm below the hollow surface) and a peristaltic pump. Both soil cores and porewater were stored on ice and shipped overnight to the University of Oregon (UO) or Chapman University (CU). We began to observe water-table drawdowns (~30–50 cm below the hollow surface) in 2016 as a result of increased temperatures in experimentally manipulated enclosures[11]. We focus here only on depth increments that were anaerobic at the time of sampling.

In the laboratory, soil samples were incubated within 1 °C of in-situ temperatures within 48 h of field collection and anaerobic incubations commenced the following day. This rapid turnaround time was intended to generate depth-specific $CH_4$ and $CO_2$ production rates that were as representative of in-situ conditions as possible. Samples were slurried with a 1:1 mixture of peat and porewater collected from the same enclosure and depth in a glove box filled with a $N_2$ atmosphere (<5% $H_2$ in the presence of a palladium catalyst) to maintain anaerobic conditions. Sample bottles were then flushed with $N_2$ for 15 min to begin the incubation. Headspace samples were analyzed over the course of 8 days (days 2, 4, 6, and 8) for $CH_4$ and $CO_2$ simultaneously using an SRI gas chromatograph equipped with a methanizer and flame ionization detector. Total $CH_4$ and $CO_2$ were calculated using Henry's Law and adjusting for solubility, temperature, and

$pH^{36}$. Methane and $CO_2$ production rates were calculated using the linear accumulation ($r^2 \geq 0.83$ in all cases) of gasses through time.

**Porewater chemistry and isotopic composition.** During WEW, porewater samples were collected 1–4 times per year during the growing season for analysis of di-hydrogen ($H_2$), acetate, $CH_4$, $CO_2$, and $DI^{14}C$ using 1.25-cm diameter PVC piezometers permanently installed at 25, 50, 75, 100, 150, and 200 cm below the hollow surface within each experimental enclosure. The 1.25-cm diameter opening was parallel to the sampling depth with a screen covering to prevent solid intrusion. Piezometers were pumped out and allowed to passively refill for 24 h prior to sampling. Given the length of the piezometers and small cross-sectional area (1.25 cm diameter), little exchange with the atmosphere was expected over 24 h. A peristaltic pump and flexible sections of silicon tubing were used to collect pore-water samples from piezometers, while surface water samples were collected using perforated stainless-steel tubes that were inserted into the peat to 10 cm or the top of the water table, whichever was shallowest.

Di-hydrogen samples were collected four times during the growing season in 2016. Immediately following collection, porewater was stored in pre-evacuated glass vials sealed with butyl stoppers, and phosphoric acid was added to each sample to preserve for shipment to the UO. At the UO, headspace samples were analyzed for $H_2$ using a Peak Performer gas chromatograph with a reducing compound photometer. The level of detection for $H_2$ was 1 ppm $mL^{-1}$.

Acetate samples were collected 1–4 times during the growing season in 2015, 2016, and 2018. Samples were filtered through a Whatman Grade GF/D glass microfiber pre-filter and a Whatman Grade GF/F glass microfiber filter, stored in 30 mL polycarbonate bottles, and immediately frozen on dry ice. Samples were shipped to CU for analysis. At CU, samples were further filtered through a Whatman 0.2-µm filter in the laboratory. Acetate concentrations were determined using a Dionex ICS-2100 ion chromatograph fitted with an AS11-HC column and AG11-HC guard column, EGC III KOH eluent generator, and ASRS 300 suppressor. Samples were neutralized using 0.1 M NaOH prior to analysis (relative standard deviation <5%) using a KOH gradient from 1 mM to 50 mM for optimal peak separation. Level of detection for acetate was 0.1 mg $L^{-1}$.

For analysis of $CH_4$ and $CO_2$ concentrations and isotopic composition, porewater was immediately filtered to 0.7 µm in the field using Whatman glass-fiber filters, then stored in pre-evacuated glass vials sealed with butyl stoppers. Phosphoric acid was added to each sample to preserve for shipment to Florida State University (FSU). Samples were analyzed for $CH_4$ and $CO_2$ concentrations and stable isotopic composition ($\delta^{13}C$) on a ThermoFinnigan Delta-V Isotope Ratio Mass Spectrometer using the headspace equilibration method with He. Each sample was analyzed twice, and the average results for each sample were recorded. Analytical precision was 0.2‰ for $^{13}C$.

Preparation of $\Delta^{14}C$-DIC samples was done at FSU by He stripping and cryogenic purification, and the resultant pure $CO_2$ was transferred to 6 mm tubes for $\Delta^{14}C$ analysis at the National Ocean Sciences Accelerator Mass Spectrometry Facility. $CO_2$ was prepared as graphite targets, and analyzed by accelerator mass spectrometry[37]. Values are reported according to the $\Delta$ notation put forth in ref. [38]. The $\Delta$ notation normalizes the radiocarbon content of a sample to a nominal $\delta^{13}C$ value (−25‰) and the collection time. The scale is linear and starts at −1000‰ when a sample has essentially 0% modern carbon, which would represent petroleum residue[39]. Analytical precision was 2‰ for $^{14}C$.

**Methanogen abundance**. Following the same protocol that was used throughout the DPH experimental phase[22], soil cores were collected in parallel with $CH_4$ and $CO_2$ production measurements, 1–2 times per year during the growing season, from 20 to 30, 40 to 50, 75 to 100, 150 to 175-cm depth increments. Soil cores were immediately frozen on dry ice, shipped to the Georgia Institute of Technology (GT), and stored at −80 °C until analysis. The total DNA was extracted from homogenized peat samples using the DNeasy PowerSoil Kit (Qiagen, formerly MoBio PowerSoil DNA extraction kit) as previously described[22,29]. The abundance of *mcrA* gene was targeted to assess the methanogen population using primer pairs ME3F, ME2R[40]. All quantitative polymerase chain reaction (qPCR) assays were performed in triplicate on a StepOnePlus platform (Applied Biosystems StepOne Plus) with PowerUp SYBR Green Master Mix (Applied Biosystems StepOne Plus). To estimate the size of the methanogenic community, the tenfold serial dilution was constructed that ranged from $10^2$ to $10^7$ molecules of the standard. pGEM-T Easy plasmid (Promega) containing *mcrA* gene fragment from *Methanococcus maripaludis* S2 was used to obtain the standard curve. In addition, a no-template DNA control was run on each qPCR plate to exclude or detect any possible contamination. Finally, the melting curve and gel electrophoresis analyses were performed to confirm the specificity of the qPCR reaction. The abundance of *mcrA* gene copy numbers was calculated and presented as gene copy numbers per dry gram of peat.

**Water-table position**. Water-table absolute elevations (a.m.s.l.) were determined at the center of all instrumented enclosures using a TruTrack Water Height Probe (Model WT-VO 2000) suspended in a 5-cm diameter stainless-steel screen well (Driller Service, Inc.—DSI) installed to an approximate depth of 1.7 m. Surveyed well cap heights were the basis for water heights along the 2000-mm probe. These absolute elevations were further reconciled to enclosure-mean hollow heights to yield water-table levels above (positive) or below (negative) the mean hollow height. Cross-enclosure comparisons of hollow-relative water-table levels demanded further adjustments for calibration shifts using direct manual observations of the presence of surface water or measured subsurface water levels.

**Surface $CH_4$ and $CO_2$ flux**. A community-level flux measurement system[41,42] was used to simultaneously measure $CO_2$ and $CH_4$ exchange at a spatial resolution that allowed for the inclusion of a representative sample of the aboveground community, including hummocks and hollows (with the exception of the widely spaced tree canopy). Briefly, we paired open-path sensors for the characterization of changing $CO_2$ and $CH_4$ concentrations within a sealed and nontransparent enclosure. A 1.2 m internal diameter flux collar was permanently placed in the bog embedded 10 to 20 cm into the surface peat to achieve an air-tight seal at ground level. Because of variable hummock-hollow topography, the volume of the collared portion of the enclosures ranged in volume from 0.45 to 0.69 $m^3$.

Measurements of $CH_4$ and $CO_2$ emissions were made 6–8 times per year during 2015–2018. The transportable and nontransparent enclosure (1.25 $m^3$) included fans for mixing, while no fans were located within the collar area to limit surface boundary layer disturbance. Total enclosed volume during measurements was around 1.8 $m^3$. We used open-path $CO_2 \times H_2O$ (LI-7500A; LiCor Inc., Lincoln, NB) and $CH_4$ (LI-7700) infrared sensors[41,42].

**Statistical analyses**. General linear mixed-effect models were used to determine the effects of depth, temperature, and elevated $CO_2$ concentrations on gas production ($CH_4$ and $CO_2$ production and $CO_2$:$CH_4$ ratios), porewater concentration ($H_2$, acetate, $CH_4$, and $CO_2$ concentrations and $CO_2$:$CH_4$ ratios), microbial (methanogen abundance), and gas emission ($CH_4$ and $CO_2$ emissions and $CO_2$:$CH_4$ ratios) data sets. In all cases, enclosure was treated as a random effect, and all other predictor variables were analyzed as fixed effects. If significant differences among depths were detected ($p < 0.05$), pairwise comparisons using Tukey's honest significant difference test ($p < 0.05$) were conducted. If not significantly different, depths were combined for linear regression analysis. In addition, stepwise multiple linear regression with Akaike Information Criterion (AIC) as the model selection condition was used to assess the ability of temperature, water-table position, and exposure to elevated $CO_2$ conditions to predict peatland $CH_4$ emissions. Data were tested for normality and log-transformed where the transformation resulted in an improvement in overall distribution. The above statistical analyses were completed using R 3.2.2 Statistical Software.

The LOESS (locally estimated scatterplot smoothing) regression for the radiocarbon plots was accomplished using a weighted least square and 2nd degree polynomial model via the smooth function in MATLAB 2017b (MathWorks, Inc).

## Data availability
Data sets pertaining to this study are in the online project archive at https://mnspruce.ornl.gov and the long-term storage in the U.S. Department of Energy's Environmental Systems Science Data Infrastructure for a Virtual Ecosystem (ESS-DIVE; https://ess-dive.lbl.gov).

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

## Acknowledgements

This material is based upon work supported by the U. S. Department of Energy, Office of Science, Office of Biological and Environmental Research. Oak Ridge National Laboratory is managed by UT-Battelle, LLC, for the U. S. Department of Energy. Funding was provided by the U. S. Department of Energy under DE-SC0014416, DE-SC00008092, DE-AC05-00OR22725, DE-SC0007144 and DE-SC0012088. Radiocarbon samples were run at the National Ocean Sciences Accelerator Mass Spectrometry Facility (Falmouth, MA). We thank Samantha Bosman and Claire Wilson for preparing and running isotope samples, Laura McCullough and Jessica Rush for assisting in running anaerobic laboratory incubations and porewater chemistry analyses, and Laurel Pfeifer-Meister for her input on laboratory-based experimental design.

## Author contributions

A.M.H. and R.M.W. wrote the paper with contributions from all co-authors. A.M.H., C.A.Z., J.K.K., and S.D.B. designed the incubation experiments. A.M.H. and C.A.Z. conducted incubations, and A.M.H. analyzed the resultant data with J.K.K. and S.D.B. A.M.H., C.A.Z., J.K.K., and S.D.B. collected porewater, and A.M.H. and C.A.Z. processed porewater for acetate and $H_2$. A.M.H. interpreted acetate and $H_2$ data with J.K.K. and S.D.B. R.M.W. and J.P.C. collected and analyzed porewater data for dissolved $CO_2$, $CH_4$, and radiocarbon signatures. M.K. and J.K. designed, collected, and analyzed the microbial data. P.J.H. collected and analyzed surface $CH_4$ flux, temperature and water-table measurements. R.M.W. completed statistical analysis of radiocarbon data, and A.M.H. completed statistical analyses of all other data sets.

## Competing interests

The authors declare no competing interests.

## Additional information

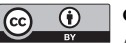 

