## [Peer Review File · Nature Communications]

Reviewers' comments:

Reviewer #1 (Remarks to the Author):

This is an impressive study investigating the vulnerability of peatlands to rising temperatures. I find the results very convincing, with a number of far-reaching implications.

This work exemplifies the considerable value of long-term studies that extend beyond the traditional three-year grant programs favoured by so many funding agencies today. The delayed response of peat at depth in their response to elevated temperatures is an absolutely important observation, something long-suspected but few studies have demonstrated. While the ^{14}C measurements of the evolved CO_2 suggest this is most likely the result of older (early Holocene) peat decomposing. It is a shame that the ^{14}C of CH_4 was not routinely measured but I agree with their comment that the radiocarbon content of CO_2 and CH_4 are likely tightly coupled in such environments. The authors provide some important observations on what might be driving the increase and delay in response to warming. Interestingly, the abundance of methanogens did not appear to increase with temperature at depth, raising the worrying possibility that the optimum temperature for such communities may be considerably higher than present in many environments and therefore may act as a strong positive feedback under future warming.

This is an important body of work that is of interest to a wide readership and provides considerable insights that urgently need investigating in future studies on carbon stores around the world. I strongly support publication.

Some minor requests:

1. In the extended data, can the authors plot the ^{14}C measurements obtained on CH_4 alongside the CO_2 where they have the data? This would help support their inference that older deeper carbon is indeed being decomposed.
2. The deep-peat heating system is impressively powerful approach. But for the imposed temperature gradients in the soil, I couldn't see anywhere about how realistic these are compared to the natural environment. Can the authors make a statement to state how realistic the temperature gradient and absolute values are under future warming scenarios?
3. Following on from the above can the authors state how their temperature regimes compare to future warming scenarios in Minnesota e.g. RCP-XX by 2100? When published I'm sure many groups would want to know so it would be good to get this in the paper.
4. I would also like to see a statement at the end of the main text stating this is a single (important) study from one specific environment but making the call/stressing the importance of similar scale studies in other environments (e.g. tropics) to investigate the impact of warming globally on other carbon stocks.

Reviewer #2 (Remarks to the Author):

Review of Hopple et al. Massive peatland carbon banks vulnerable to rising temperatures.

This is a good paper. It is another paper resulting from the SPRUCE temperature and CO_2 manipulation on a northern peatland. It is relevant and should have wide interest given the potential importance to biospheric – climate feedbacks. Some results from this experiment were reported several years ago but this paper deals with six years of results. The manuscripts contain new results that show the response after a number of years is different than the first two years. The results are worthy of publication and would be a good addition to the literature.

I think it is important that the authors begin to quantify the changes they observed and in the context of the emerging peatland literature on climate change and ecosystem resilience.

The SPRUCE experiment applies large (some might say huge) temperature change – far greater than what would be expected in peatlands due to the observed and projecting warming in the 2100 century. The authors' lowest temperature manipulation is probably in excess of what peatlands are likely to experience, especially when you consider they are discussing below-ground processes. Peatlands contain a lot of water and this makes them inherently thermally conservative. It takes a lot of joules to raise a peatland 20C – some of the manipulations are +90C. The heat capacity and thermal conductivity of peat will inhibit such extreme changes. It also means the responses around +20C are the most interesting. This does not mean the results are not useful, but the question is can the authors place their results in this context. In the plots of the CH₄ and CO₂ response, the temperature axis is the values at which the incubations were done – I think? The relevant question is how much different is the temperature of the incubations from their natural temperature? As the authors are well aware there is a huge debate over the mobilization of carbon at depth in soils. I think back to the discussions a number of years ago over temperature responses and that discussion still continues. For example see Ågren, G. I. and E. Bosatta (2002). "Reconciling differences in predictions of temperature response of soil organic matter." *Soil Biology and Biochemistry* 34(1): 129-132, response to the debate between Gairdina and Ryan and Davidson et al. in *Nature* (2000). Ågren argued that material exposed to different temperatures than its 'native' temperature is a transient response. Since the authors saw different results over the six years are they still observing the transient response (see Luo, Y. and E. Weng (2011). "Dynamic disequilibrium of the terrestrial carbon cycle under global change." *Trends in Ecology & Evolution* 26(2): 96-104, and the literature that follows from this review). In the SPRUCE experiment, the system is being exposed to a significant change, which produces a large disequilibrium. How do you know the time course of the response signal? In all ecosystem manipulations one would expect the biggest bang for the buck early and then the signal should decrease. The rate of response is a function of the mass of C in the system and the changes in input. We know peatlands have been subject to climate variations over millennia and they seemed to have adjusted. Is there sufficient resilience in these systems that you have observed a blip on the transient curve as the system is an attempt to remain within its range of control (see Morris, P. J., A. J. Baird, D. M. Young and G. T. Swindles (2015). "Untangling climate signals from autogenic changes in long-term peatland development." *Geophysical Research Letters* 42(24): 10788-10797.)

Nigel Roulet, McGill University, November 2019

Reviewer #3 (Remarks to the Author):

This manuscript presents results related to anaerobic decomposition in the SPRUCE experiment where large mesocosms have been exposed to deep peat warming, surface warming and elevated atmospheric CO₂. Belowground decomposition processes were investigated with in vitro measurements of CO₂ and CH₄ production, methanogen community composition, pore water chemistry and isotopic composition. Surface CO₂ and CH₄ fluxes were also measured. The authors conclude that deep peat stocks are vulnerable to warming following a short lag period and that the ecosystem is shifting to become more methanogenic, as supported by a decrease in CO₂/CH₄ ratios in production, porewater and surface flux under higher temperatures. While the conclusions are supported by the data to some extent, I think more caution is needed in the interpretation. This may be partially solved by expanding the text and adding a more fulsome discussion of the results. I assume that this was originally written as a letter for one of the other Nature journals, but for *Nature Communications* could be expanded, particularly given the large amount of data included in the paper.

Specifically, although trends are shown in the figures that support the authors' conclusions, for many of the trendlines, I remain unconvinced that the trend is truly representing the data. Because of this, the relative strong conclusions made in the text, start to come into question when the data itself is scrutinized. I do recognize the incredible value of the SPRUCE experiment and the unique opportunity to study controls on peatland C cycling in this set-up and so I encourage the authors to tell a more

nuanced story that more clearly acknowledges the high level of variability in the data and potential that the trends observed are maybe not as “troubling” as they lay out in the text. I give specific comments below about this and other aspects of the design and text to consider in a revision.

Lines 57-59: In the study set up, deep peat warming was introduced prior to surface warming, while in reality, rising surface temperatures will eventually warm deeper peat, but with a lag following surface warming (so the opposite of the treatment). It would be nice to see at least a sentence acknowledging this and how it may impact the results observed (e.g., would the lag be even longer in reality since you first have a lag to warm the deep peat and then a biogeochemical lag?)

Line 80: I would argue that DOC is also a decomposition product (or at least intermediate)

Lines 99-102: have you looked at all at the syntrophic community? I wonder if changes in the supply of substrate to the methanogenic community are contributing to greater efficiency in CH₄ production. Beyond the scope of this paper I guess.

Lines 310-311: Since only anaerobic depths were investigated in the CO₂ and CH₄ production experiments I think this has the potential to overestimate the shifts in CO₂/CH₄ production ratio. If now there are larger aerobic sections of the profile, CH₄ production in those zones would be low, results in high CO₂/CH₄ production ratios that should balance out or exceed shifts in the anaerobic zone. In any case, it is not always clear in the main body of the text that only anaerobic production is discussed, so check this throughout.

Line 328: specify what component ¹⁴C is being measured on – I think it is only the DIC.

Line 328: specify the slotted opening length on the piezometer. Also, were they sealed in some way so that atmospheric air could not affect pore water chemistry?

Line 336: Write out UO in full.

Line 357: I suggest you move the precision for ¹⁴C to the following paragraph where you talk about the ¹⁴C analytical method.

Line 376: should be methanogenic community

Lines 391-393: I’m not really sure what you mean here. Why not just calibrate directly for each enclosure based on the manual measurement?

Lines 401-403: Were both hummocks and hollows included in the collars used for fluxes? If this varied markedly it could have a large impact on the measured flux.

Lines 405-408: How frequently were the fluxes measured over the study period? This starts to come out in the figures, but should also be stated here. I assume that these collars had intact plant communities so that CO₂ emissions also include plant respiration. Is that correct and if so, how does it affect your interpretation of the results? At least for the CO₂/CH₄ ratio, I would argue that it is more important to know the proportion of net CO₂ uptake that is lost as CH₄, since even increases in CH₄ uptake could be more than balanced by large net C storage when considering radiative forcing.

Figure 1: First, thanks to the authors for showing all the data point clearly. Both the 30 cm and 50 cm plots appear to me to be largely driven by one point at low temperature. Once you move to higher temperatures, the scatter is so large that I have a hard time imagining that the trendlines would be significant. Did you test these without the low values? Then, for the 75 -200 cm plot, again, I found the trends unconvincing. For 200 cm points, the trend looks flat, while for 125, it might even look opposite the main trendline shown. I’m not arguing that the statistical analysis is incorrect, I’m just

wondering how much it is really convincing given the large scatter in the data. My biggest concern is that the text as currently written does not reflect that this amount of scatter exists and that definitely needs a fuller discussion.

Figure 2: Super interesting. My only question is how much these patterns can be extended to CH₄ (as noted on line 86). The authors provide a reference for the fact that ¹⁴C in DIC and CH₄ are tightly coupled, but can shifts in processes linked to warming affect this relationship?

Figure 4: in the middle panel (b), I cannot see how the 10 cm trendline really represents the data well. There are several points at high temperatures with some of the highest CO₂/CH₄ porewater ratios, so I am not convince that the general pattern at the surface is a deep decline in the ratio. Given that deeper water table was also often observed, this high CO₂ concentration in the near surface actually makes more sense to me. Again, I think this fuller discussion of the data, warts and all, is really needed in the text.

In the (c) panel, the trends are more convincing, but I find it hard to see which points line up with the trends line. Can you use color instead of symbols here? For panel (a), this is where I wonder how including the aerobic depths would change this trend. For clarity, I suggest adding anaerobic to the y-axis title of this panel.

Extended data figure 3: I suggest removing the trendline since it's not significant

Extended data figure 6, 7: In general, I have the same comment about these as many of the main figures. On first look, I see a gunshot blast of data without a really obvious trend. I note that log axes are used, so that the changes along the trend line are actually quite large, but in that case, so is the scatter. I'm just not sure how much faith I have in these trends.

Extended data figure 9: If this also includes plant respiration, then I wonder how much of the trend is drive by increases in that component given that the plant productivity has increased with warming.

Extended data figure 10: I suggest that this is not needed. The water table differences between the treatments is shown clearly in extended data figure 4.

Maria Strack

We appreciate the extensive comments provided by the reviewers, as well as their overall positive appraisal of our manuscript. Below, we have provided complete responses to reviewer comments, along with references to the revised manuscript text when applicable.

Reviewers' comments:

Reviewer #1 (Remarks to the Author):

This is an impressive study investigating the vulnerability of peatlands to rising temperatures. I find the results very convincing, with a number of far-reaching implications.

This work exemplifies the considerable value of long-term studies that extend beyond the traditional three-year grant programs favoured by so many funding agencies today. The delayed response of peat at depth in their response to elevated temperatures is an absolutely important observation, something long-suspected but few studies have demonstrated. While the ^{14}C measurements of the evolved CO_2 suggest this is most likely the result of older (early Holocene) peat decomposing. It is a shame that the ^{14}C of CH_4 was not routinely measured but I agree with their comment that the radiocarbon content of CO_2 and CH_4 are likely tightly coupled in such environments. The authors provide some important observations on what might be driving the increase and delay in response to warming. Interestingly, the abundance of methanogens did not appear to increase with temperature at depth, raising the worrying possibility that the optimum temperature for such communities may be considerably higher than present in many environments and therefore may act as a strong positive feedback under future warming.

This is an important body of work that is of interest to a wide readership and provides considerable insights that urgently need investigating in future studies on carbon stores around the world. I strongly support publication.

Some minor requests:

1. In the extended data, can the authors plot the ^{14}C measurements obtained on CH_4 alongside the CO_2 where they have the data? This would help support their inference that older deeper carbon is indeed being decomposed.

We have added an extended data figure showing the $^{14}\text{CH}_4$ alongside the DI^{14}C for comparison (listed as Extended Data Figure 3).

2. The deep-peat heating system is impressively powerful approach. But for the imposed temperature gradients in the soil, I couldn't see anywhere about how realistic these are compared to the natural environment. Can the authors make a statement to state how realistic the temperature gradient and absolute values are under future warming scenarios?

We appreciate reviewer 1 and 2's comments that we need to address how our experimental design aligns with real world climate change scenarios. In response to these concerns, we would like to clarify that deep peat will naturally warm in parallel with surface warming due to the propagation of heat downwards through the peat column-albeit with a small temporal offset. Per Hanson et al. (2011) "Often overlooked in warming studies is the reality that deep soil temperatures will also become elevated as they equilibrate with new annual temperatures (Baxter, 1997; Hu & Feng, 2003; Song et al., 2019). Such deep warming has not been achieved using previous warming technologies. Instead, preferential heating of surface soils has occurred as soils deeper in the profile remain unaffected."

The SPRUCE study approach allows for such warming, as well as the investigation of complex and non-linear peatland-atmosphere interactions that may mitigate or amplify anthropogenic climate change. For example, by increasing temperatures to +9 °C above ambient, we can (1) explore ecosystem-wide response surface thresholds and (2) employ non-linear curve fitting that may prove vital to understanding the response of peatlands to warmer temperatures (Amthor et al., 2010). These analyses would not be possible without the inclusion of multiple temperature treatments in a regression-based experimental design.

Based on current greenhouse gas emission rates (Friedlingstein et al., 2019), forcing estimates in line or greater than the RCP 8.5 scenario are likely to occur (IPCC, 2013), resulting in 2.6 – 4.8 °C increases in global mean surface temperatures by 2100. However, both models and observations indicate that northern boreal forests and tundra will continue to be exposed to greater warming than most other terrestrial biomes. This is largely due to positive feedback effects related to decreases in surface albedo resulting from decreasing sea ice and shorter periods of winter snow cover. The effects of temperature increases are further compounded by the heightened frequency of extreme heat events which are expected to expose peatlands to acute heat stress, exceeding the conditions for which vegetation is currently adapted. Thus, under a globally averaged projection of +4 °C warming by the end of this century, boreal regions could experience temperature increases up to 8.3 ± 1.9 °C (Schadel et al., 2016; Gauthier et al., 2015).

We have revised the text to emphasize these points in LINES 58-75:

“Capturing comprehensive peatland responses to climate change requires the use of active surface and deep soil warming because deep soil temperatures will naturally increase in concert with rising annual air temperatures (Song et al., 2019; Hanson et al., 2011; Hu & Feng, 2003). Based on current greenhouse gas emission rates (Friedlingstein et al., 2019), forcing estimates in line or greater than the RCP 8.5 scenario are likely to occur (Stocker et al., 2013). This will result in 2.6 – 4.8 °C increases in global mean surface temperatures by 2100, with Minnesota experiencing 4.07 – 4.14 °C increases in annual air temperatures (Jiang et al., 2017). However, both models and observations indicate that northern boreal forests and tundra will continue to be exposed to greater warming than most other terrestrial biomes. This is largely due to positive feedback effects related to decreases in surface albedo resulting from decreasing sea ice and shorter periods of winter snow cover (Stocker et al., 2013). The effects of temperature increases are further compounded by the heightened frequency of extreme heat events which are expected to expose peatlands to acute heat stress, exceeding the conditions for which vegetation is currently adapted. Thus, under a globally averaged projection of +4 °C warming by the end of this century, boreal regions may experience temperature increases up to 8.3 ± 1.9 °C (Schadel et al., 2016; Gauthier et al., 2015); thus, the +9 °C treatment implemented in this study represents an upper limit on what can be expected under the most extreme scenarios. Additionally, implementing warming treatments at a range of temperature allows for the exploration of ecosystem-wide response surface thresholds and non-linear curve response fitting (Eppinga et al., 2009).”

Hanson, Paul J., et al. "A method for experimental heating of intact soil profiles for application to climate change experiments." *Global Change Biology* 17.2 (2011): 1083-1096.

Baxter, D. O. "A comparison of deep soil temperature: Tennessee versus other locations." *Transactions of the ASAE* 40.3 (1997): 727-738.

Hu, Qi, and Song Feng. "A daily soil temperature dataset and soil temperature climatology of the contiguous United States." *Journal of applied meteorology* 42.8 (2003): 1139-1156.

Soong, J. L., Phillips, C. L., Ledna, C., Koven, C. D., & Torn, M. S. (2019). CMIP5 models predict rapid and deep soil warming over the 21st century. *Journal of Geophysical Research: Biogeosciences*, e2019JG005266.

Aronson, Emma L., and Steven G. McNulty. "Appropriate experimental ecosystem warming methods by ecosystem, objective, and practicality." *Agricultural and Forest Meteorology* 149.11 (2009): 1791-1799.

Friedlingstein, Pierre, et al. "Global carbon budget 2019." *Earth System Science Data* 11.4 (2019): 1783-1838.

Stocker, T. F., Qin, D., Plattner, G. K., Tignor, M., Allen, S. K., Boschung, J., ... & Midgley, P. M. (2013). Climate change 2013: The physical science basis. *Contribution of Working Group I to the Fifth Assessment Report of the Intergovernmental Panel on Climate Change*. Cambridge University Press, Cambridge, United Kingdom and New York, NY, USA.

Jiang, L., Fang, X., & Chen, G. (2017). Refuge Lake Reclassification in 620 Minnesota Cisco Lakes under Future Climate Scenarios. *Water*, 9(9), 675.

Schädel, C., Bader, M. K. F., Schuur, E. A., Biasi, C., Bracho, R., Čapek, P., ... & Graham, D. E. (2016). Potential carbon emissions dominated by carbon dioxide from thawed permafrost soils. *Nature Climate Change*, 6(10), 950.

Gauthier, S., Bernier, P., Kuuluvainen, T., Shvidenko, A. Z., & Schepaschenko, D. G. (2015). Boreal forest health and global change. *Science*, 349(6250), 819-822.

Eppinga, M. B., Rietkerk, M., Wassen, M. J., & De Ruiter, P. C. (2009). Linking habitat modification to catastrophic shifts and vegetation patterns in bogs. *Plant Ecology*, 200(1), 53-68.

3. Following on from the above can the authors state how their temperature regimes compare to future warming scenarios in Minnesota e.g. RCP-XX by 2100? When published I'm sure many groups would want to know so it would be good to get this in the paper.

Agreed, we have included temperature projections specific to Minnesota and a discussion of how this relates to our treatments in Lines 58-75.

4. I would also like to see a statement at the end of the main text stating this is a single (important) study from one specific environment but making the call/stressing the importance of similar scale studies in other environments (e.g. tropics) to investigate the impact of warming globally on other carbon stocks.

We agree that this is also an important caveat to our study and have added the following text to Lines 182-188:

“It should be noted that we report the warming response of one bog and CH₄ emissions and anaerobic C cycling have been shown to vary substantially among southern boreal habitats (Hanson et al., 2016; Zalman et al., 2018). Thus, the results described here do not necessarily reflect the expected or observed responses from other peatland habitats. Nonetheless, our conclusions have far-reaching implications for predicting ecosystem-atmosphere feedbacks that exist in systems experiencing climate change, highlighting the need for similar manipulative studies implemented across a diverse array of ecosystem types and biomes.”

Hanson, P. J., Gill, A. L., Xu, X., Phillips, J. R., Weston, D. J., Kolka, R. K., ... & Hook, L. A. (2016). Intermediate-scale community-level flux of CO₂ and CH₄ in a Minnesota peatland: putting the SPRUCE project in a global context. *Biogeochemistry*, 129(3), 255-272.

Zalman, C., Keller, J. K., Tfaily, M., Kolton, M., Pfeifer-Meister, L., Wilson, R. M., ... & Finzi, A. (2018). Small differences in ombrotrophy control regional-scale variation in methane cycling among Sphagnum-dominated peatlands. *Biogeochemistry*, 139(2), 155-177.

Reviewer #2 (Remarks to the Author):

Review of Hopple et al. Massive peatland carbon banks vulnerable to rising temperatures.

This is a good paper. It is another paper resulting from the SPRUCE temperature and CO₂ manipulation on a northern peatland. It is relevant and should have wide interest given the potential importance to biospheric – climate feedbacks. Some results from this experiment were reported several years ago but this paper deals with six years of results. The manuscripts contain new results that show the response after a number of years is different than the first two years. The results are worthy of publication and would be a good addition to the literature.

I think it is important that the authors begin to quantify the changes they observed and in the context of the emerging peatland literature on climate change and ecosystem resilience.

The SPRUCE experiment applies large (some might say huge) temperature change – far greater than what would be expected in peatlands due to the observed and projecting warming in the 2100 century. The authors' lowest temperature manipulation is probably in excess of what peatlands are likely to experience, especially when you consider they are discussing below-ground processes. Peatlands contain a lot of water and this makes them inherently thermally conservative. It takes a lot of joules to raise a peatland 2°C – some of the manipulations are +9°C. The heat capacity and thermal conductivity of peat will inhibit such extreme changes. It also means the responses around +2°C are the most interesting. This does not mean the results are not useful, but the question is can the authors place their results in this context.

Please see our above response to Point 2 made by Reviewer 1.

In the plots of the CH₄ and CO₂ response, the temperature axis is the values at which the incubations were done – I think? The relevant question is how much different is the temperature of the incubations from their natural temperature? As the authors are well aware there is a huge debate over the mobilization of carbon at depth in soils. I think back to the discussions a number of years ago over temperature responses and that discussion still continues. For example see Ågren, G. I. and E. Bosatta (2002). "Reconciling differences in predictions of temperature response of soil organic matter." *Soil Biology and Biochemistry* 34(1): 129-132, response to the debate between Gairdina and Ryan and Davidson et al. in *Nature* (2000).

We agree with Reviewer 2 that conducting laboratory incubations at temperatures outside of the native temperature range of the ecosystem can result in soil organic matter decomposition dynamics that differ from incubations done within the native temperature range. However, we would like to clarify that our incubations were done with 1 °C of the in-situ, depth-specific soil temperatures. We refer to the following Lines 581-584:

“Positive CH₄ production temperature responses from peat samples collected (a) 30 cm, (b) 50 cm, and (c) 75-200 cm below the hollow surface at S1 Bog and anaerobically incubated within 1 °C of the in-situ temperatures.”

And Lines 382-385:

“In the laboratory, soil samples were incubated within 1 °C of in-situ temperatures within 48 hours of field collection and anaerobic incubations commenced the following day. This rapid turnaround time was intended to generate depth-specific CH₄ and CO₂ production rates that were as representative of in-situ conditions as possible.”

We hope this text clarifies that the incubation temperatures reflect those experienced under field conditions.

Since the authors saw different results over the six years are they still observing the transient response (see Luo, Y. and E. Weng (2011). "Dynamic disequilibrium of the terrestrial carbon cycle under global change." *Trends in Ecology & Evolution* 26(2): 96-104, and the literature that follows from this review). In the SPRUCE experiment, the system is being exposed to a significant change, which produces a large disequilibrium. How do you know the time course of the response signal? In all ecosystem manipulations one would expect the biggest bang for the buck early and then the signal should decrease. The rate of response is a function of the mass of C in the system and the changes in input. We know peatlands have been subject to climate variations over millennia and they seemed to have adjusted. Is there sufficient resilience in these systems that you have observed a blip on the transient curve as the system is an attempt to remain within its range of control (see Morris, P. J., A. J. Baird, D. M. Young and G. T. Swindles (2015). "Untangling climate signals from autogenic changes in long-term peatland development." *Geophysical Research Letters* 42(24): 10788-10797.)

Evidence for temperature acclimation in individual site-based studies is mixed, but a recent global synthesis of global soil warming experiments showed little indication of temperature acclimation in short-term warming studies (Carey et al. 2016). From Carey et al. (2016): “Thus, our data provide limited evidence of acclimation of soil respiration to experimental warming in several major biome types, contrary to the results from multiple single-site studies.” In fact, we have observed the opposite of acclimation in deep peat at SPRUCE with a significant lag period before any effects of warming on CH₄ production. The second part of the reviewer’s comment appears to focus on the net C balance of the ecosystem, and our study only focuses on the respiration part of this balance, and more specifically on CH₄, and thus the long-term response of the C stocks of peatlands to climate change is a separate question. Interestingly, results from SPRUCE indicate that warming is causing a relatively large loss of C from the system (Hanson et al., submitted). However, we do acknowledge that we are unsure of the longer-term C responses from SPRUCE in Lines 188-191:

“Finally, as the SPRUCE experiment continues into the next decade, it remains to be seen whether the observed temperature effects will persist, diminish through acclimation of the ecosystem (Carey et al., 2016; Bridgham et al., 2008), or be further modified by the impacts of changes in water-table position and eCO₂ on plant community productivity and composition.”

Carey, J. C., J. Tang, P. H. Templer, K. D. Kroeger, T. W. Crowther, A. J. Burton, J. S. Dukes, B. Emmett, S. D. Frey, M. A. Hessel, L. Jiang, M. B. Machmuller, J. Mohan, A. M. Panetta, P. B. Reich, S. Reinsch, X. Wang, S. D. Allison, C. Bamminger, S. Bridgham, S. L. Collins, G. de Dato, W. C. Eddy, B. J. Enquist, M. Estiarte, J. Harte, A. Henderson, B. R. Johnson, K. S. Larsen, Y. Luo, S. Marhan, J. M. Melillo, J. Peuelas, L. Pfeifer-Meister, C. Poll, E. Rastetter, A. B. Reinmann, L. L. Reynolds, I. K. Schmidt, G. R. Shaver, A. L. Strong, V. Suseela, and A. Tietema. 2016. Temperature response of soil respiration largely unaltered with experimental warming. *Proceedings of the National Academy of Sciences of the United States of America* 113:13797-13802.

Bridgham, S. D., Pastor, J., Dewey, B., Weltzin, J. F., & Updegraff, K. (2008). Rapid carbon response of peatlands to climate change. *Ecology*. 89(11), 3041-3048.

Nigel Roulet, McGill University, November 2019

Reviewer #3 (Remarks to the Author):

This manuscript presents results related to anaerobic decomposition in the SPRUCE experiment where large mesocosms have been exposed to deep peat warming, surface warming and elevated atmospheric CO₂. Belowground decomposition processes were investigated with in vitro measurements of CO₂ and CH₄ production, methanogen community composition, pore water chemistry and isotopic composition. Surface CO₂ and CH₄ fluxes were also measured. The authors conclude that deep peat stocks are vulnerable to warming following a short lag period and that the ecosystem is shifting to become more methanogenic, as supported by a decrease in CO₂/CH₄ ratios in production, porewater and surface flux under higher temperatures. While the conclusions are supported by the data to some extent, I think more caution is needed in the interpretation. This may be partially solved by expanding the text and adding a more fulsome discussion of the results. I assume that this was originally written as letter for one of the other Nature journals, but for Nature Communications could be expanded, particularly given the large amount of data included in the paper.

Specifically, although trends are shown in the figures that support the authors conclusions, for many of the trendlines, I remain unconvinced that the trend is truly representing the data. Because of this, the relative strong conclusions made in the text, start to come into question when the data itself is scrutinized. I do recognize the incredible value of the SPRUCE experiment and the unique opportunity to study controls on peatland C cycling in this set-up and so I encourage the authors to tell a more nuanced story that more clearly acknowledges the high level of variability in the data and potential that the trends observed are maybe not as “troubling” as they lay out in the text. I give specific comments below about this and other aspects of the design and text to consider in a revision.

We appreciate Reviewer 3’s detailed comments, as well as her concerns that we did not acknowledge the variability of the data presented in Figures 1 and 4 and Extended Data Figures 6 and 7. The variability in Figure 1 has been addressed in the manuscript in Lines 123-148. Further, it seems that Figure 4 may have been misinterpreted by this reviewer; we discuss this in more detail below. Finally, we have revised our text to reflect the variation in Extended Data Figures 6 and 7, but also point out the utility of the inferences we made when studying ecological cascades that span multiple scales and years. We provide our full responses below.

Lines 57-59: In the study set up, deep peat warming was introduced prior to surface warming, while in reality, rising surface temperatures will eventually warm deeper peat, but with a lag following surface warming (so the opposite of the treatment). It would be nice to see at least a sentence acknowledging this and how it may impact the results observed (e.g., would the lag be even longer in reality since you first have a lag to warm the deep peat and then a biogeochemical lag?)

We have acknowledged this possibility in Lines 92-93:

“It is possible that this lag period may be longer under natural conditions where the soil profile warms from the top down.”

Line 80: I would argue that DOC is also a decomposition product (or at least intermediate)

While we acknowledge that DOC can act as an intermediate in decomposition, several studies have shown that DOC acts as a primary decomposition source at this site, as well as in other ombrotrophic bogs (Chanton et al., 2008; Corbett et al., 2013; McFarlane et al., 2018; Tfaily et al., 2014; Hopple et al., 2019). This been demonstrated by strong overlap in ¹⁴C signatures of DOC and respiration products and the stark difference in ¹⁴C signatures of DOC and solid peat.

Chanton, J. P., Glaser, P. H., Chasar, L. S., Burdige, D. J., Hines, M. E., Siegel, D. I., ... & Cooper, W. T. (2008). Radiocarbon evidence for the importance of surface vegetation on fermentation and methanogenesis in contrasting types of boreal peatlands. *Global Biogeochemical Cycles*. 22(4).

Corbett, J. E., Burdige, D. J., Tfaily, M. M., Dial, A. R., Cooper, W. T., Glaser, P. H., & Chanton, J. P. (2013). Surface production fuels deep heterotrophic respiration in northern peatlands. *Global Biogeochemical Cycles*. 27(4), 1163-1174.

McFarlane, K.J., Hanson P.J., Iversen C.M., Phillips J.R., & Brice D.J. (2018) Local spatial heterogeneity of Holocene carbon accumulation throughout the peat profile of an ombrotrophic Northern Minnesota bog. *Radiocarbon*. 60, 941-962.

Tfaily, M. M., Cooper, W. T., Kostka, J. E., Chanton, P. R., Schadt, C. W., Hanson, P. J., ... & Chanton, J. P. (2014). Organic matter transformation in the peat column at Marcell Experimental Forest: humification and vertical stratification. *Journal of Geophysical Research: Biogeosciences*. 119(4), 661-675.

Hopple, A. M., Pfeifer-Meister, L., Zalman, C. A., Keller, J. K., Tfaily, M. M., Wilson, R. M., ... & Bridgman, S. D. (2019). Does dissolved organic matter or solid peat fuel anaerobic respiration in peatlands?. *Geoderma*, 349, 79-87.

Lines 99-102: have you looked at all at the syntrophic community? I wonder if changes in the supply of substrate to the methanogenic community are contributing to greater efficiency in CH₄ production. Beyond the scope of this paper I guess.

While assessing changes in the syntrophic microbial community is an important aspect of understanding ecosystem response to climate change, it was beyond the scope of this paper. However, our group has indeed investigated the community composition and relative abundance of known syntrophs in soils of the S1 bog from 2015 to 2018. Similar to what we have reported here for methanogen abundance, we do not observe statistically significant changes in the relative abundance of syntrophs at the DNA sequence level. Thus, at this time, we hypothesize that we are observing a physiological response of the microbial community to changing environmental conditions, rather than a change in growth or community composition. However, we are indeed seeing greater efficiency in CH₄ production as indicated by the CO₂:CH₄ ratios shown in Figure 4.

Lines 310-311: Since only anaerobic depths were investigated in the CO₂ and CH₄ production experiments I think this has the potential to overestimate the shifts in CO₂/CH₄ production ratio. If now there are larger aerobic sections of the profile, CH₄ production in those zones would be low, results in high CO₂/CH₄ production ratios that should balance out or exceed shifts in the anaerobic zone. In any case, it is not always clear in the main body of the text that only anaerobic production is discussed, so check this throughout.

We agree with the reviewer that basing our assertion that the ecosystem has shifted into a more methanogenic state solely on CO₂:CH₄ production ratios measured from anaerobic soil depths would potentially overestimate shifts in this ratio, as well as our overall conclusions. For this

reason, we paired three independent lines of evidence that support our claims: decreases in the CO₂:CH₄ ratios of (1) production, (2) porewater, and (3) emissions. While the production ratios were measured only at anaerobic depths, the porewater ratios measured from 10-200 cm below the soil surface (depths less than ~30 cm were often wet, but above the water table and likely included aerobic processes). Similarly, the emission measurements (which were taken at the soil surface and thus represent the net balance of aerobic and anaerobic processes) also show decreases in CO₂:CH₄ ratios with increasing temperature.

This is noted in Lines 150-154:

“Increases in CH₄ and CO₂ production rates throughout the soil profile resulted in exponential and linear increases in peatland CH₄ (p < 0.0001; Figure 3) and dark CO₂ (p < 0.0001; Extended Data Figure 10) emissions, respectively. Exponential increases in CH₄ emissions were also observed during DPH⁶ and have persisted over the past four years despite general decreases in water-table position with warming (Extended Data Figure 5).”

As well as Lines 172-175:

“While porewater concentration and emission measurements encompass both autotrophic and heterotrophic respiration, we found that the declining CO₂:CH₄ ratios in both the porewater and the emissions were caused by increasingly methanogenic heterotrophic soil respiration (rather than decreasing CO₂ rates).”

Line 328: specify what component 14C is being measured on – I think it is only the DIC.

We have revised this text to “DI¹⁴C”.

Line 328: specify the slotted opening length on the piezometer. Also, were they sealed in some way so that atmospheric air could not affect pore water chemistry?

The following text was added to Lines 398-402 to clarify the piezometer design:

“The 1.25 cm diameter opening was parallel to the sampling depth with a screen covering to prevent solid intrusion. Piezometers were pumped out and allowed to passively refill for 24 hours prior to sampling. Given the length of the piezometers and small cross-sectional area (1.25 cm diameter) little exchange with the atmosphere was expected over 24 hours.”

Line 336: Write out UO in full.

We have fully written out this abbreviation at its first use (Line 376).

Line 357: I suggest you move the precision for 14C to the following paragraph where you talk about the 14C analytical method.

We have moved this text as the reviewer requested.

Line 376: should be methanogenic community

We have made this revision.

Lines 391-393: I'm not really sure what you mean here. Why not just calibrate directly for each enclosure based on the manual measurement?

Because of high within plot variability of hummock hollow elevations and variation in the overall elevation of the S1 Bog within which the SPRUCE experimental plots are distributed, it was necessary to normalize the plot data to enable plot-to-plot comparisons of water table elevation with respect to hollow height across treatments. The plot measurements are located at the center of each treatment plot/enclosure and the hollow height at that location may not be a perfect representation of the mean hollow height for that plot. Therefore, adjustments were made for the single-point absolute elevation data to better represent the mean hollow for each plot. This normalized data set enabled cross treatment comparisons.

Lines 401-403: Were both hummocks and hollows included in the collars used for fluxes? If this varied markedly it could have a large impact on the measured flux.

The large collars were placed and chosen to include approximately half hollow and half hummock coverage. These amounts did not differ markedly. We have noted this in Lines 471-472.

Lines 405-408: How frequently were the fluxes measured over the study period? This starts to come out in the figures, but should also be stated here. I assume that these collars had intact plant communities so that CO₂ emissions also include plant respiration. Is that correct and if so, how does it affect your interpretation of the results? At least for the CO₂/CH₄ ratio, I would argue that it is more important to know the proportion of net CO₂ uptake that is lost as CH₄, since even increases in CH₄ uptake could be more than balanced by large net C storage when considering radiative forcing.

We have added the following text to Lines 479 to denote how frequently the fluxes were measured:

“Measurements of CH₄ and CO₂ fluxes were made 6-8 times per year during 2015-2018.”

Yes, the collars contain intact plant communities as stated in Lines 469-472:

“A community-level flux measurement system was used to simultaneously measure CO₂ and CH₄ exchange at a spatial resolution that allowed for the inclusion of a representative sample of the aboveground community (with the exception of the widely spaced tree canopy).”

We include our interpretation of how autotrophic and heterotrophic respiration influence the observed CO₂:CH₄ ratios in Lines 164-175:

“During WEW, CO₂:CH₄ production ratios decreased exponentially with warmer temperatures across all depth increments ($p < 0.0001$; Figure 4a). Similarly, porewater CO₂:CH₄ ratios linearly decreased with warming ($p < 0.0001$; Figure 4b), although the severity of this effect decreased with depth ($p < 0.001$). There was higher variability in the shallowest (10 cm) depth as evidence by the larger spread of 95% confidence intervals which likely resulted from fluctuations in the water table depth in the near surface. Additionally, while emission CO₂:CH₄ ratios were stable during the first two years of WEW (2015 and 2016), this ratio began to decrease in 2017 ($p < 0.01$), with an even steeper decrease observed in 2018 ($p < 0.001$; Figure 4c). While porewater concentration and emission measurements encompass both autotrophic and heterotrophic respiration, we found that the declining CO₂:CH₄ ratios in both the porewater and the emissions were caused by increasingly methanogenic heterotrophic soil respiration (rather than decreasing CO₂ rates).”

Figure 1: First, thanks to the authors for showing all the data point clearly. Both the 30 cm and 50 cm

plots appear to me to be largely driven by one point at low temperature. Once you move to higher temperatures, the scatter is so large that I have a hard time imagining that the trendlines would be significant. Did you test these without the low values? Then, for the 75 -200 cm plot, again, I found the trends unconvincing. For 200 cm points, the trend looks flat, while for 125, it might even look opposite the main trendline shown. I'm not arguing that the statistical analysis is incorrect, I'm just wondering how much it is really convincing given the large scatter in the data. My biggest concern is that the text as currently written does not reflect that this amount of scatter exists and that definitely needs a fuller discussion.

While we appreciate the reviewer concerns over the variability of the production data, we disagree that this variation is not adequately addressed in the text. Per the reviewer's own appreciative comments, our figures clearly show the variation in CH₄ and CO₂ production temperature responses separated by statistically different depth increments, as well as the associated r² and p values for each linear regression. Methane and CO₂ production rates were calculated using the linear accumulation of gasses through time, with r² ≥ 0.83 in all cases. Thus, we are confident that all data points included in the figures are reliable production rates. However, we do acknowledge the reviewers concerns that temperature responses may be driven by a single low point in the 30 and 50 cm depth increments. When these low values are dropped, the temperature responses remain significant (p ≤ 0.05). We would also like to point out that the reviewer should not interpret the 75-200 cm temperature responses individually. These deeper depths are not statistically different and, thus, are aggregated to assess temperature responses.

Furthermore, a large section of the main text (~300 words) is devoted to interpreting the high variation in CH₄ production following whole-ecosystem warming in Lines 125-148:

“However, temperature explained much less of the variation in surface CH₄ production during WEW (r² = 0.07, Fig. 1) relative to DPH (r² = 0.71). Climate-induced perturbations to the ecosystem, such as changes in water-table, increased below-ground exudation of labile plant compounds, or changes in plant and/or microbial community composition, likely had cascading ecological effects on peatland CH₄ production, muting the direct effects of temperature on this process in surface soils. For example, water-table drawdowns observed under WEW (Extended Data Figure 5) likely oxidized terminal electron acceptors and intermittently decreased the soil anaerobic zone, suppressing rates of surface methanogenesis and potentially stimulating CH₄ oxidation. Conversely, two years of exposure to eCO₂ stimulated rates of CH₄ production in surface soils (Extended Data Figure 6), possibly through increased availability of the methanogenic substrates, acetate and H₂, due to enhanced plant root exudation. We observed decreases in these substrates throughout the entire soil profile during WEW (Extended Data Figures 7 and 8), indicating heightened microbial substrate utilization with warming. Additionally, this negative temperature response became stronger for surface acetate concentrations following the introduction of eCO₂ (p < 0.05; Extended Data Figure 9). These results suggest that warming and eCO₂ drove increases in surface CH₄ production through kinetic- and substrate-based effects that enhanced rates of methanogen activity, overwhelming any suppression effects associated with water-table fluctuations.”

Figure 2: Super interesting. My only question is how much these patterns can be extended to CH₄ (as noted on line 86). The authors provide a reference for the fact that 14C in DIC and CH₄ are tightly coupled, but can shifts in processes linked to warming affect this relationship?

The tight coupling between CO₂ and CH₄ radiocarbon values have been demonstrated in a wide range of wetlands spanning different temperature regimes. An analysis of several combined previous studies shows this 1:1 relationship is quite strong and robust $r^2 = 0.97$, $n = 67$ (Chanton et al., 2008, *GBC* 22:GB4022). This empirical relationship is also consistent with a theoretical explanation of why this should occur. Peatland CH₄ is produced via CO₂ reduction or acetate disproportionation. If methane is produced via CO₂ reduction, we expect the methane and CO₂ to have the same radiocarbon signature since the fractionation associated with methanogenesis is accounted for by correcting both to a common $\delta^{13}\text{C}$ when values are reported as $\delta^{14}\text{C}$ as we have done in this paper. If CH₄ is being produced from acetate disproportionation, then CH₄ should have the same radiocarbon value as the CO₂ since both are being produced from the 2 C of the same initial molecule (acetate). Thus, observation and theory agree that CH₄ and CO₂ radiocarbon values are coupled.

Figure 4: in the middle panel (b), I cannot see how the 10 cm trendline really represents the data well. There are several points at high temperatures with some of the highest CO₂/CH₄ porewater ratios, so I am not convinced that the general pattern at the surface is a deep decline in the ratio. Given that deeper water table was also often observed, this high CO₂ concentration in the near surface actually makes more sense to me. Again, I think this fuller discussion of the data, warts and all, is really needed in the text. In the (c) panel, the trends are more convincing, but I find it hard to see which points line up with the trends line. Can you use color instead of symbols here? For panel (a), this is where I wonder how including the aerobic depths would change this trend. For clarity, I suggest adding anaerobic to the y-axis title of this panel.

Regarding the comments on panel (a): We have added “anaerobic” to the y-axis.

Regarding the comments on panel (b), we suspect that the reviewer misinterpreted the 10 cm CO₂:CH₄ ratio trend in panel (b): A closer inspection of the 10 cm trend line shows that, while a few of the CO₂:CH₄ ratios (3 points) are high at high temperatures, the majority (15 points) are low. We agree that fluctuations in water table near the surface will enhance the variability of porewater CO₂:CH₄ ratios, as evidenced by the greater spread of the 95% confidence intervals relative to those of deeper depths. We have clarified the text as follows Lines 166-170:

“Similarly, porewater CO₂:CH₄ ratios linearly decreased with warming ($p < 0.0001$; Figure 4b), although the severity of this effect decreased with depth ($p < 0.001$). There was higher variability in the shallowest (10 cm) depth as evidenced by the larger spread of 95% confidence intervals which likely resulted from fluctuations in the water table depth in the near surface.”

We disagree that the greater variation in 10 cm porewater CO₂:CH₄ ratios warrants greater discussion in the text because the trend is the same as that observed at other depths and in the accompanying panels.

Regarding comments on panel (c): We have revised the figure to use colors instead of shapes.

Extended data figure 3: I suggest removing the trendline since it's not significant

We have removed the trendline from Extended Data Figure 3.

Extended data figure 6, 7: In general, I have the same comment about these as many of the main figures. On first look, I see a gunshot blast of data without a really obvious trend. I note that log axes are used, so that the changes along the trend line are actually quite large, but in that case, so is the scatter. I'm just not

sure how much faith I have in these trends.

Connecting molecular-level variables with ecosystem functions is a challenging and notoriously difficult task. We acknowledge that temperature explains a small amount of the microbial substrate variation measured in this study and quantify this low explanatory power with r^2 values. However, given the large amount of data collected over multiple years, the trends are statistically significant and, we argue, ecologically meaningful. As the reviewer requested, we have revised our text to point out the high variation in microbial substrate concentration, but also point out the merits of combining these multiscale analyses when examining ecosystem responses in Lines 140-145:

“While temperature explains a small, but statistically significant proportion of the variance in methanogenic substrate concentrations (acetate: $r^2 = 0.02-0.17$, $p < 0.001$ to 0.08 ; H_2 : $r^2 = 0.02$, $p < 0.01$), our ability to connect rapidly utilized, low-molecular weight microbial substrates with process rates and, ultimately, ecosystem function remains a powerful approach for understanding ecological cascades.”

Extended data figure 9: If this also includes plant respiration, then I wonder how much of the trend is drive by increases in that component given that the plant productivity has increased with warming.

Our dark CO_2 fluxes encompass both autotrophic and heterotrophic respiration. We do not include measurements of plant productivity in this study and do not focus on its contribution to CO_2 flux. Instead our focus is understanding the peatland methane cycle and changes in the $CO_2:CH_4$ ratio. We include Extended Data Figure 9 (now Extended Data Figure 10) so that the reader may see the underlying change in CO_2 flux and how it influences the observed change in $CO_2:CH_4$ emission ratios.

Extended data figure 10: I suggest that this is not needed. The water table differences between the treatments is shown clearly in extended data figure 4.

We have removed Extended Data Figure 10 from the paper.

Maria Strack

REVIEWERS' COMMENTS:

Reviewer #2 (Remarks to the Author):

I have read the revised version of this manuscript and the authors have addressed the issues I raised. I think it is a good paper.

Nigel Roulet

Reviewer #3 (Remarks to the Author):

I thank the authors for their careful and detailed responses to all of the reviewer comments. It is clear that they are considered all comments in depth and incorporated changes that improve the clarity the study's potential limitations and interpretation of results.

These changes have adequately addressed my questions from the original submission and I would suggest publication in the current form.

I just note a typo for "temperature" on line 64 and that "warms" should be warm on line 93.

Maria Strack

REVIEWERS' COMMENTS:

Reviewer #2 (Remarks to the Author):

I have read the revised version of this manuscript and the authors have addressed the issues I raised. I think it is a good paper.

Nigel Roulet

Reviewer #3 (Remarks to the Author):

I thank the authors for their careful and detailed responses to all of the reviewer comments. It is clear that they are considered all comments in depth and incorporated changes that improve the clarity the study's potential limitations and interpretation of results.

These changes have adequately addressed my questions from the original submission and I would suggest publication in the current form.

I just note a typo for "temperature" on line 64 and that "warms" should be warm on line 93.

We have corrected this typo.

Maria Strack